# Analyzing Physical-Mechanical and Hydrophysical Properties of Sandy Soils Exposed to Long-Term Hydrocarbon Contamination

**Ivan Lange** *[ID], **Pavel Kotiukov and Yana Lebedeva**

Department of Hydrogeology and Engineering Geology, Saint-Petersburg Mining University,
199106 Saint-Petersburg, Russia
* Correspondence: lange_iyu@pers.spmi.ru

**Abstract:** This paper aims to investigate the issue of sandy soil contamination by oil hydrocarbons. Laboratory procedures used to study conditionally pure and contaminated sands include grain-size measurements and evaluation of physical-mechanical and hydrophysical properties. The results of the analysis of sand samples using visual and microscopic studies and sieve analysis show that, unlike in conditionally pure sands, in contaminated samples, the surface of mineral particles is covered by hydrocarbon film. The presence of the latter enables micro- and macroaggregates to be formed. Studies of the physical and hydrophysical properties of sands using a technique with pre-weighed glass containers, as well as a filtering device, SPETSGEO pipes, showed that, in comparison with conditionally pure samples, contaminated specimens of sandy soils have lower densities and higher permeabilities and water yields. Testing the mechanical properties of contaminated sands on the GPP-30 direct shear apparatus using the consolidated dried shear strength method revealed an increase in the angle of internal friction with a decrease in specific adhesion compared to conditionally clean sands.

**Keywords:** sandy soils; oil hydrocarbon contamination; particle-size distribution; hydrocarbon films; microaggregates; sandy soil properties

## 1. Introduction

Nowadays, the growth of urban conglomerates involves the use of territories that were previously considered unfit for urban development [1,2]. First of all, it relates to areas where former industrial facilities are located. These could include grounds of oil refineries and oil-processing plants, as well as tank batteries, fuel and lubricant storage, and other structures [3–5]. The operation of such facilities comprises refining, processing, storing, and management of hydrocarbons, and it inevitably entails their loss [6–9]. According to statistics, the overall loss of solid and liquid hydrocarbons may exceed 2.5 tons and 140.0 tons per year, respectively, when operating tank batteries and filling gas stations [10–12]. The influx of such an amount of hydrocarbons into the ground, in the upper part of which generally sandy soils are located, results in the transformation of their composition and physical-mechanical and hydrophysical properties. According to published works, the flow of petroleum hydrocarbons into sandy soils leads to the formation of hydrocarbon films on the surface of mineral grains and an increase in the size of sand particles. It is noted that there is a decrease in their permeability and shear strength [13–18]. However, the conducted studies show some inconsistency with the mechanism of oil pollution of sands in natural conditions. This is primarily due to the fact that the composition and properties of contaminated sandy soils were determined by mixing them with liquid hydrocarbons. Under natural conditions, this process is more complicated. The infiltration of petroleum hydrocarbons is accompanied by their transformation under the action of chemical oxidation. After a long time has passed, light hydrocarbon fractions evaporate

from the composition of liquid hydrocarbons, and heavier ones form hydrocarbon films on the surface of mineral particles. This mechanism was confirmed when studying the reasons for the failure of an oil reservoir after 20 years of operation. Leaks of petroleum hydrocarbons during the operation of the oil tank led to a deterioration in the properties of sandy soils at its base. Because of this, there was an uneven deformation of its sandy base, which caused its destruction [19]. At the same time, despite the availability of practical examples, as well as scientific papers on the effect of oil pollution on soil properties, the mechanism of transformation of the composition and properties of sandy soils compared to clay soils has not yet been sufficiently studied. Therefore, there is a need for experimental studies on sandy soils.

## 2. Research Methods and Results

In order to carry out experimental studies and comparative analysis of transformation in sandy soil composition and properties, which had been affected by hydrocarbon pollution, 12 samples of both contaminated and non-contaminated sands were taken (6 samples of each type). Sampling sites were confined to the area of an abandoned military shooting range, which was previously supported by the continuous use of oil hydrocarbons as fuel and lubricants. Prolonged use of oil products resulted in the contamination of soils which was specific to this territory. Contaminated areas in the form of concentric zones were easily identified by the dark-gray and black colors of sandy deposits and by the distinctive odor of oil hydrocarbons. During trial pit excavation, some sand intercalations containing oil hydrocarbons were found.

By visually analyzing the collected sand samples, it was established that uncontaminated sands were characterized by yellowish-beige color, medium-grained structure (prevailing grain size 0.25–0.5 mm), and quartz-feldspar composition. In contrast to conditionally pure sands, contaminated samples had a dark-brown to black color and were similar to poorly cemented sandstone with a pronounced smell of oil hydrocarbons (Figure 1).

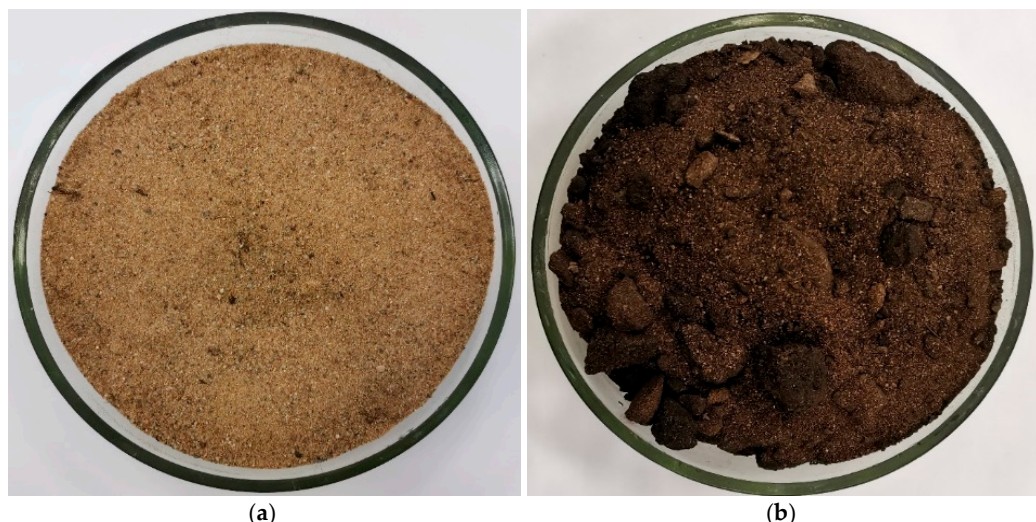

(**a**)                    (**b**)

**Figure 1.** Visual distinctions between conditionally pure (**a**) and contaminated sands (**b**).

To determine the amount of oil hydrocarbons in contaminated sandy soils a fluorometric analysis was performed using the analyzing device Fluorat 02-3M. The substance of such analysis performed according to the normative document (PND F 16.1:2.21-98) was to extract oil hydrocarbons from contaminated samples with a hexane solution and to determine their content using a chromatography column [20]. The results thus obtained showed that in test samples the oil hydrocarbon content was 17.5 g/kg.

The visual description of sand samples was complemented by a microscopic study that led to the discovery of hydrocarbon films on the surface of mineral particles in contaminated sands (Figure 2).

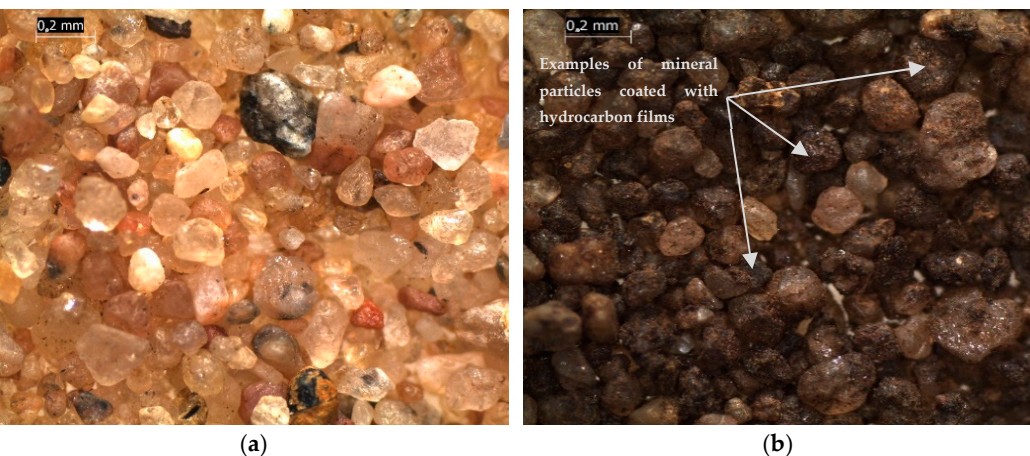

(**a**)  (**b**)

**Figure 2.** Image of sandy soils obtained using a LEICA DM750 microscope: (**a**) conditionally pure sands; (**b**) sands contaminated by oil hydrocarbons.

The presence of oxidized hydrocarbon films on the surface of mineral particles in contaminated sands led to the formation of microaggregates ranging in size from 0.5 to 5.0 mm. Detected films played the role of cementing substances (Figure 3).

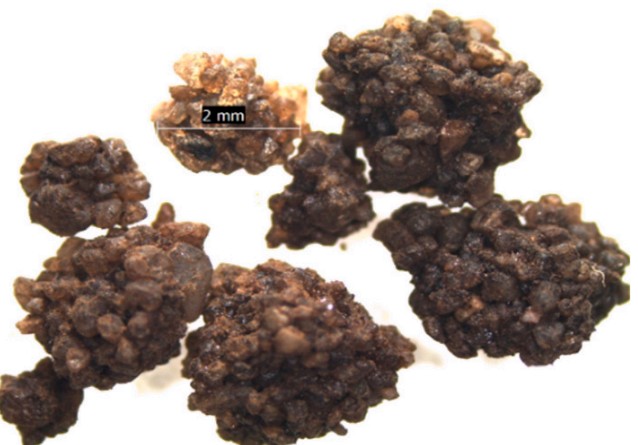

**Figure 3.** Microaggregates of sand particles contaminated by oil hydrocarbons (the image was taken with a LEICA DM750 microscope).

Studying sand grain-size distribution by means of the sieve method (GOST 12536-2014) revealed that conditionally pure samples were within the range of medium-grained sands, whereas contaminated sands were classified as coarse and very coarse-grained (Figure 4).

Analyzing the granular composition of samples under study shows that the formation of hydrocarbon films on the surface of sand grains and the building up of microaggregates results in increased coarseness. As seen in Figure 4, the content of particles with a size between 0.5 and 2.0 mm in conditionally pure sands varies from 12 to 22%, whereas in contaminated sands it exceeds 51%. Moreover, in contaminated samples, we observe the emergence of individual aggregates with sizes from 2.0 to 5.0 mm.

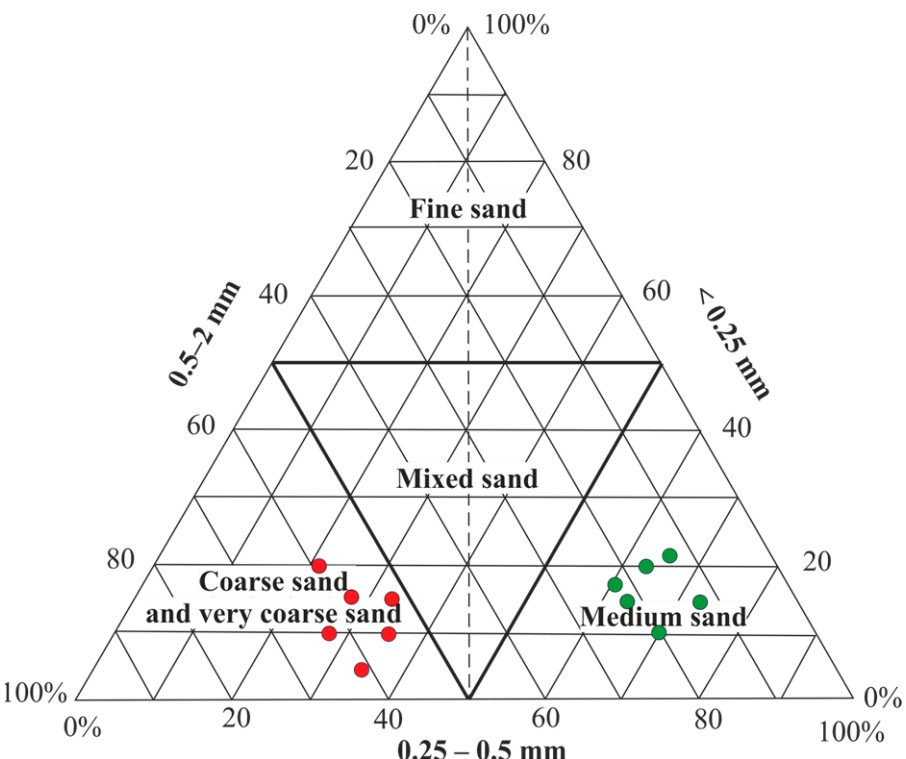

**Figure 4.** Triangular diagram of grain-size distribution for studied sands: green dots—conditionally pure sands; red dots—sands covered by hydrocarbon films.

The transformation of granular composition towards the increased content of large fractions affects the density of contaminated sands with both dense and loose structures. Measurements of sand density were based on a method, which used previously weighed glass flasks of the same volume. The loosest structure of sand samples was achieved by freely filling flasks through the funnel. The most compact structure was attained by filling the flasks with sand in small portions, which was followed by compaction using the impact of the tamper on the flask walls. The density was calculated using the following formula:

$$\rho = (m_1 - m_2)/V, \tag{1}$$

where $m_1$ is the mass of sandy soil with the loosest or densest structure, g; $m_2$ is the mass of the glass flask, g; V is the volume of sand filling up the flask, cm$^3$.

The results of performed calculations indicate that under contamination, soils have lower densities in the loosest and in densest conditions compared to conditionally pure samples (Table 1).

**Table 1.** The density measurements for the sands under study.

| Laboratory Sample | Loose Structure | | | Dense Structure | | |
|---|---|---|---|---|---|---|
| | Density, g/cm$^3$ | | Average Value [1], g/cm$^3$ | Density, g/cm$^3$ | | Average Value [1], g/cm$^3$ |
| | from | to | | From | To | |
| | Conditionally pure sand | | | | | |
| 1–6 | 1.41 | 1.43 | 1.42 | 1.71 | 1.72 | 1.72 |
| | Contaminated sand | | | | | |
| 7–12 | 1.31 | 1.34 | 1.32 | 1.51 | 1.52 | 1.51 |

[1] Calculated as a weighted average value.

The density of contaminated loose and dense sands reduces by 0.10 g/cm$^3$ and 0.21 g/cm$^3$ respectively. In percentage terms, the density of contaminated sand in a loose structure compared to conditionally pure sand decreased by 7%, and in a dense composition by 12%. This could be explained by the aggregation of mineral particles. Aggregates, which constitute mineral grains cemented by hydrocarbon films, take more space per unit of volume compared to uncontaminated particles. This has a direct impact on the porosity of sand. Previous studies investigating properties of sands under hydrocarbon contamination demonstrated that, along with the density being reduced from 1.43 down to 1.01 g/cm$^3$ and from 1.68 to 1.25 g/cm$^3$ for loose and dense samples, respectively, porosity increased from 0.47 to 0.58 and from 0.37 to 0.49, respectively [19].

The lower density and higher porosity of contaminated sands lead to a change in permeability. To quantify the water permeability of sandy soils, laboratory tests with soil specimens were carried out. According to the normative technique, the permeability coefficient was measured while maintaining a permanent hydraulic gradient throughout the entire test ("steady-state filtration condition") [21]. For this purpose, a permeameter—a filtration tube designed by SPECGEO—was used.

The results acquired allowed the analysis of the relationship between the filtration coefficient and sand density. To perform such an experiment, specimens of sandy soils with the loosest and densest structures were prepared. Once the necessary sand density was reached and the apparatus was filled with distilled water, permeability tests were started. Over the course of the experiment, the time needed to filter a given volume of water was measured, while the head gradient was maintained at the level of 1.0. A total of 24 tests were completed—12 for each type of sand. The results acquired were used to develop a plot relating the filtration coefficient (at T = 10 °C) and sand density (Figure 5).

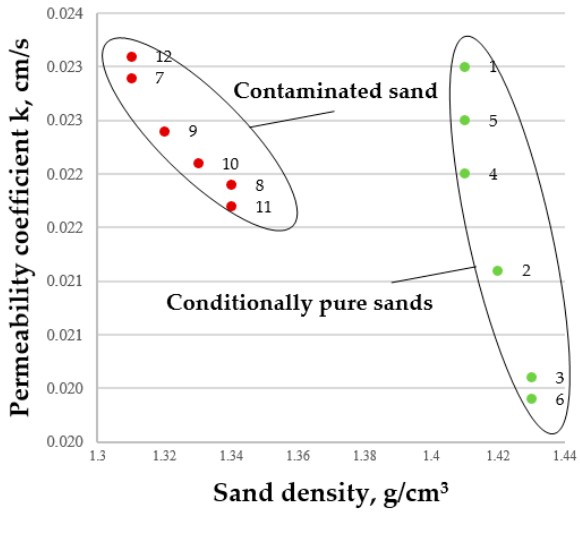

(**a**)

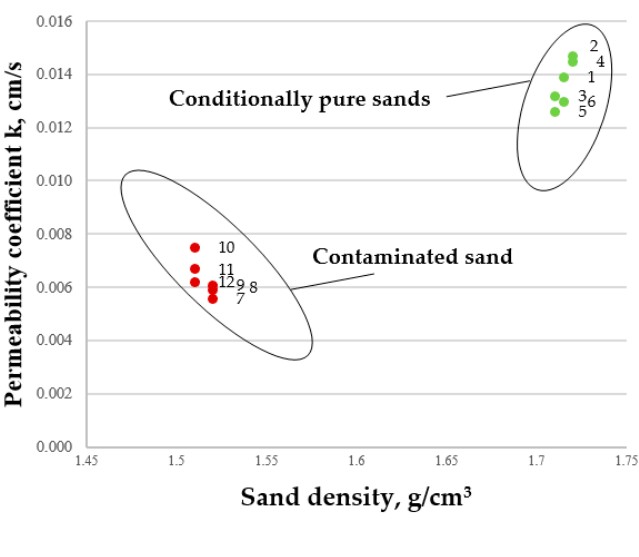

(**b**)

**Figure 5.** Scatter plots of the permeability coefficient for conditionally pure and contaminated sands versus their density: (**a**) loose structure; (**b**) dense structure.

Figure 5 shows that, despite differences in loose sand density, contaminated specimens had almost comparable permeability relative to conditionally pure sands. The filtration coefficient of contaminated sands was about 18.7 ÷ 20.0 m/day compared to 17.3 ÷ 19.8 m/day for pure sands. In the case of dense specimens, the permeability appeared to be significantly different. The filtration coefficient for contaminated sands ranged from 5.1 to 6.5 m/day, for pure sands, it varied from 11.2 to 13.0 m/day.

Laboratory experiments with sandy soils were complemented by strength tests performed using the direct shear apparatus GPP-30, designed by Institute "Hydroproject". For this purpose, the consolidated drained method of shear strength testing was used. It

implies shearing previously consolidated samples in compression at a slow rate, which allows pore pressure built up during the experiment to dissipate. The method is widely used for the determination of a sand's shear strength [22–26].

For experiments to be conducted, 12 air-dried samples of pure and contaminated sands were prepared—6 of both loose and dense types of structures. In accordance with the recommendations outlined in normative documents, a set of three samples was subjected to controlled normal stress (σ)—100 kPa, 150 kPa, and 200 kPa in each experiment. After that, inducing strain lateral load was applied until samples failed. Thus, peak stresses (τ) were found (Table 2).

**Table 2.** Shear strength test results for sands.

| Laboratory Sample | Conditionally Pure Sand | | Laboratory Sample | Contaminated Sand | |
| | Stress, kPa | | | Stress, kPa | |
| | Normal (Confining), σ | Shear (Tangential), τ | | Normal (Confining), σ | Shear (Tangential), τ |
|---|---|---|---|---|---|
| | | | Loose structure | | |
| 1 | 100 | 65 | 7 | 100 | 75 |
| 2 | 150 | 99 | 8 | 150 | 119 |
| 3 | 200 | 130 | 9 | 200 | 170 |
| Laboratory sample | Conditionally Pure Sand Stress, kPa | | Laboratory sample | Contaminated Sand Stress, kPa | |
| | Normal (Confining), σ | Shear (Tangential), τ | | Normal (Confining), σ | Shear (Tangential), τ |
| | | | Dense structure | | |
| 4 | 100 | 95 | 10 | 100 | 85 |
| 5 | 150 | 126 | 11 | 150 | 126 |
| 6 | 200 | 160 | 12 | 200 | 190 |

Test results produced graphs of shear stress as a function of normal stress (Figure 6). They were used to determine shear strength parameters—cohesion c (kPa) and angle of internal friction ϕ (deg.).

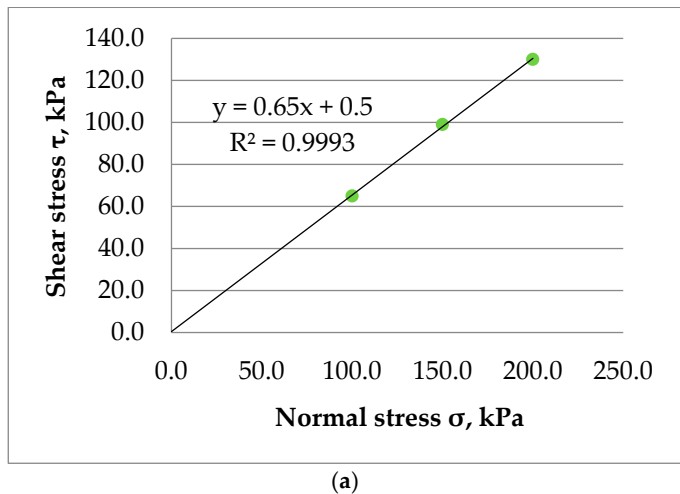

(**a**)

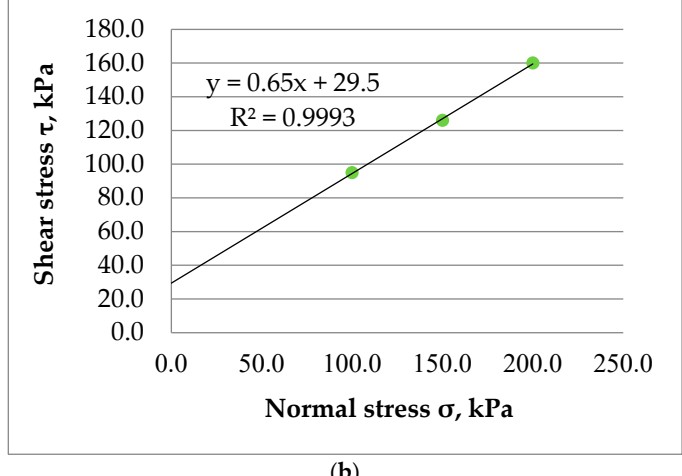

(**b**)

**Figure 6.** *Cont*.

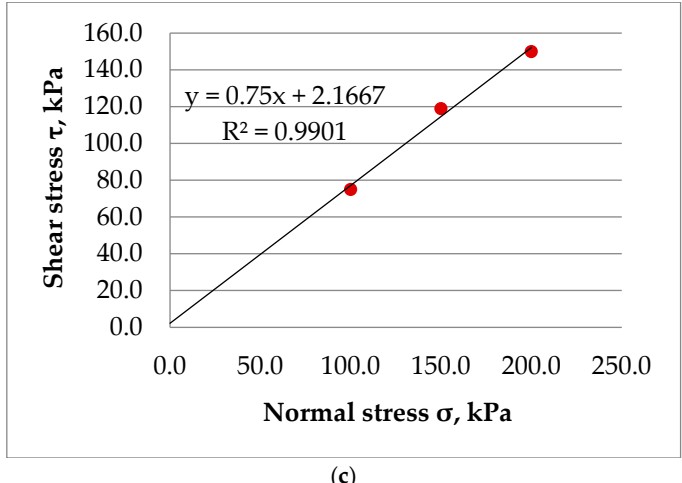
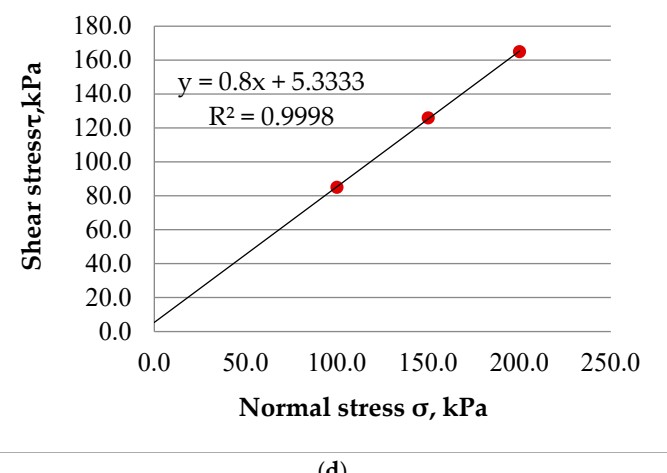

(**c**)　　　　　　　　　　　　　　　　　　　　　　　(**d**)

**Figure 6.** Graphical representation of shear strength factors for conditionally pure sands with loose (**a**) and dense (**b**) structures; contaminated sands with loose (**c**) and dense (**d**) structures: green dots—conditionally pure sands; red dots—sands covered by hydrocarbon films.

Shear strength parameters were determined using a straight-line equation describing the linear curve fitting the test result points. The equation is also known as Coulomb's shear strength equation:

$$\tau = \sigma \cdot \tan \phi + c, \tag{2}$$

where $\sigma$ is an effective normal stress on the rupture plane, $kg/cm^2$; $\tan \phi$ is termed as the coefficient of internal friction of the soil; $\phi$ is the angle of internal friction; the intercept $c$ is the cohesion of the soil, kPa.

Figure 6 shows that in conditionally pure loose sand (Figure 6a), $\phi$ is 33° and the c value is at a near-zero level. For dense sand (Figure 6b), $\phi$ remains the same, whereas c increases significantly up to 30 kPa. In contaminated sands, another pattern is detected. An observation can be made that $\phi$ of sand with loose structure (Figure 6c) is 37°, c = 2 kPa. In the case of a dense structure, $\phi$ increases by 2° (39°), and c increases to 5 kPa. The results suggest that the increase in the density of contaminated sands has less impact on change in strength parameters compared to conditionally pure sands.

In studying the strength of sands, special attention was given to the deformation behavior and its duration under different normal stresses (Figures 7 and 8).

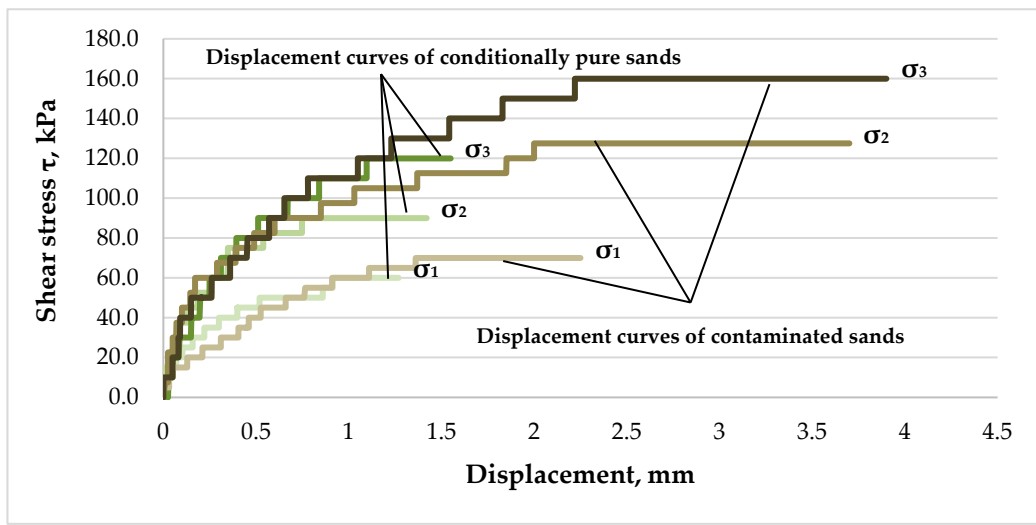

**Figure 7.** Shear stress–displacement curves for loose sands: $\sigma_1$, $\sigma_2$, and $\sigma_3$—normal stress values, 100, 150, and 200 kPa, respectively.

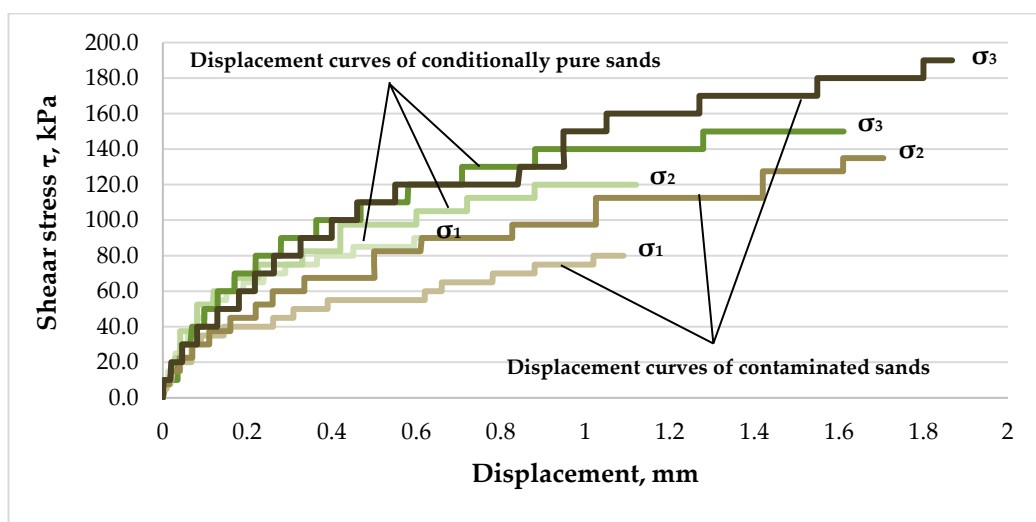

**Figure 8.** Shear stress–displacement curves for dense sands: $\sigma_1$, $\sigma_2$, and $\sigma_3$—normal stress values, 100, 150, and 200 kPa, respectively.

The resulting curves allowed tracing certain patterns—shear displacements in contaminated sands were developing for a longer period of time than such in conditionally pure sands under the same normal load. Depending on normal load value and sand density, the overall duration of shear deformation in conditionally pure sands did not exceed 2–3 h, whereas in contaminated samples it could last up to 5–10 h.

## 3. Discussion

Analyzing the results of laboratory studies of conditionally pure and contaminated sands revealed some patterns in changes in the sandy soil grain-size distribution and their physical-mechanical and hydrophysical properties that occurred under long-term hydrocarbon contamination. Detected patterns can be shown in the form the following scheme (Figure 9).

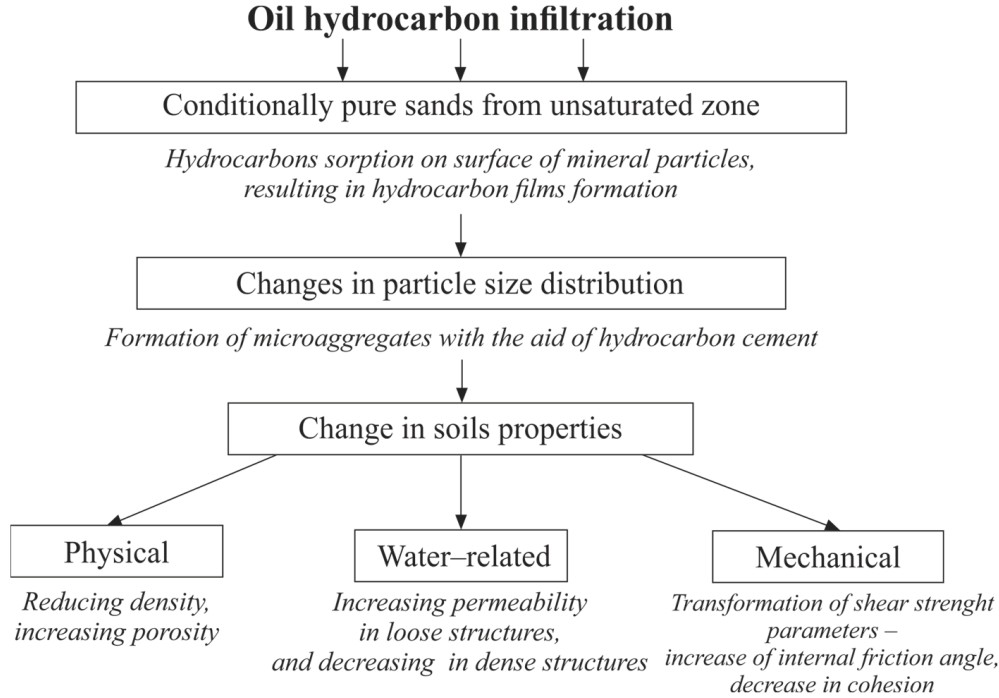

**Figure 9.** Scheme of sandy soil transformation under long-lasting hydrocarbon contamination.

It is seen from Figure 9 that input of petroleum hydrocarbons in the upper part of the profile, namely the unsaturated zone built of air-dried sands, is accompanied by a sorption process [27–29]. According to published data, the sorption capacity of soils is measured by their granulometric size composition [30–33]. The highest capacity, in relation to hydrocarbons, is typical of clay soils, which are able to sorb and retain more than 40 L/m$^3$. The lowest capacity is characteristic of coarse-grained sands, which is less than 8 L/m$^3$ [19,34]. However, the nature of this phenomenon is quite complex. It is defined by two main factors: the formation of polar groups within oil hydrocarbons and the presence of unlike charges on the surfaces of sand particles [35].

It is known that when oil hydrocarbons interact with air oxygen their active transformation generated by chemical oxidation occurs [36,37]. Such a process results in the evaporation of light hydrocarbon fractions and in a growing number of polar groups (–OH,–COOH, etc.) within heavier hydrocarbons, promoting an increase in their activity rate [38,39]. Functional groups composing oil hydrocarbons can be regarded as polyelectrolytes, which can be sorbed at the surfaces with unlike charges. Such sorption is considered to be irreversible [40,41]. The number of detained hydrocarbon molecules depends on their composition, molar mass, sorption energy, and the properties of mineral grain surfaces in contact (Figure 10) [13].

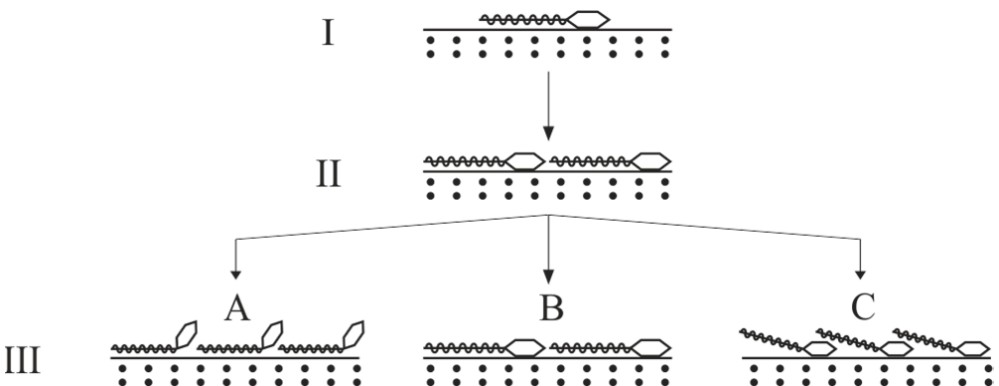

**Figure 10.** Some sorption mechanisms of oil hydrocarbons with polar groups, displaying alignment of molecules at the surface of mineral particles [13]: I–III—successive stages of sorption; A–C correspond to weak, intermediate, and strong interactions.

The present understanding of unlike charges forming on the surface of sand mineral particles, based on spectrophotometric studies, is that it relates to the charge heterogeneity, which is expressed by active center formation (Lewis acids and bases). Silicon atoms with unlike charges play the role of such centers, which may chemically react with adsorbed hydrocarbon molecules (Figure 11) [13].

As implied above, it follows that the interaction of sandy soils with oil hydrocarbons is a complex process, the development of which depends on the sands' mineral and grain-size composition, as well as on the hydrocarbons' structure. It is worth noting that, in order to reveal mechanisms of oil hydrocarbon sorption on the surface of mineral particles, individual thorough studies are required. However, despite the insufficient knowledge of such interactions nature, results of the investigations carried out revealed some relationships between changes in contaminated soil composition and transformation of their physical-mechanical properties. Summing up the results, the following conclusions can be drawn.

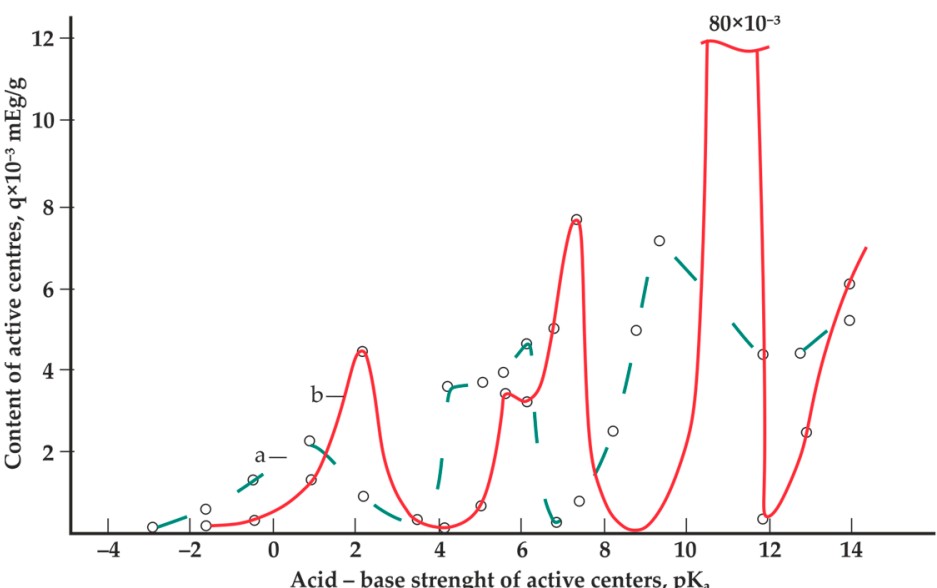

**Figure 11.** Distribution curve of acid-base centers on the surface of natural (**a**) and enriched (**b**) quartz sand [13].

## 4. Conclusions

Petroleum hydrocarbon loss in the operation of industrial facilities results in the contamination of sandy soils, thus adversely impacting their composition and properties. The origin of such changes lies in the physical-mechanical interaction between polar groups of oil hydrocarbons and active centers of sand grains. The result of this interaction consists in the sorption of oil hydrocarbons by sandy soils, in the process of which changes in grain-size distribution and hence physical-mechanical and hydrophysical properties of soils occur. To establish regularities in sand transformation under hydrocarbon contamination, laboratory studies were performed. Summing up the results, it can be concluded that:

(1) Oil hydrocarbon sorption causes the formation of hydrocarbon films on the grain surface of sandy soils and promotes the development of microaggregates, ranging in size from 0.5 to 5.0 mm, which are cemented by detected films.

(2) The content of microaggregates in contaminated sands increases their coarseness by over 51% up to coarse-grained and very coarse-grained types compared to conditionally pure sands, which are classified as medium-grained.

(3) Aggregated state of mineral grains in contaminated sands affects their density. In loose and dense structures, it becomes lower by 0.10 and 0.21 g/cm$^3$, respectively, in contrast to conditionally pure sands, which have densities of 1.42 g/cm$^3$ and 1.72 g/cm$^3$, respectively.

(4) Changes in the density of contaminated sands have an impact on their permeability. In loose sands, the filtration coefficient is similar to that of conditionally pure samples, varying within the range of 18.7–20.0 m/day. The filtration coefficient of dense sands decreases to 5.1–6.5 m/day.

(5) Studying strength parameters of contaminated sands using the consolidated drain method of the direct shear test shows that, in comparison with conditionally pure sands, contaminated samples have higher internal friction angle φ (up to 39°) and display greater cohesion c (up to 5 kPa). Particular attention should be drawn to the duration of the contaminated sands' shear deformations—depending on the applied normal stress and sample density, the experiment can last up to 10 h.

**Author Contributions:** Conceptualization, I.L.; methodology, Y.L.; validation, P.K.; formal analysis, I.L.; investigation, I.L.; resources, Y.L.; writing—original draft preparation, I.L.; visualization, P.K. All authors have read and agreed to the published version of the manuscript.

**Funding:** This research received no external funding.

**Institutional Review Board Statement:** Not applicable.

**Informed Consent Statement:** Not applicable.

**Data Availability Statement:** The data will be provided when the reader asked for it.

**Conflicts of Interest:** The authors declare no conflict of interest.

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
