# Peer review of "Analyzing Physical-Mechanical and Hydrophysical Properties of Sandy Soils Exposed to Long-Term Hydrocarbon Contamination"

_sustainability, doi:10.3390/su15043599_

Round 1
Reviewer 1 Report
1. Petroleum hydrocarbons loss in the operation of industrial facilities results in contamination of soils, thus impacting their composition and properties. The authors carry out experimental studies and get the following results: in comparison with conditionally pure sands, contaminated ones have lower density, higher permeability, higher internal friction angle along with smaller cohesion.
2. Because the sorption capacity of coarse-grained sands is much smaller than that of clay soils, reviewed literature about the physical-mechanical and hydrophysical properties of contaminated clay soils is strongly suggested to be complemented and discuss in this study.
Reviewer 2 Report
This manuscript provides a physical-mechanical comparison of contaminated and uncontaminated sands through indoor tests. The results indicate that contaminated specimens of sandy soils have lower density, higher permeability and water return rate. Furthermore, testing mechanical properties of contaminated sands reveals increase in internal friction angle along with decrease in specific cohesion as compared to uncontaminated sands. However, the following issues need to be explained in detail:
(1) In the introduction, it is suggested to describe research status in detail and combine with engineering cases, so as to be able to highlight the significance of the research. You can refer to the following four papers:
Influence of cyclic wetting-drying on the shear strength of limestone with a soft interlayer.
Displacement prediction of Jiuxianping landslide using gated recurrent unit (GRU) networks.
Stability analysis of the reservoir bank landslide with weak interlayer considering the influence of multiple factors.
Evolution trend of the Huangyanwo rock mass under the action of reservoir water fluctuation.
A comprehensive review of machine learning‐based methods in landslide susceptibility mapping
(2) In section 2, whether the conditionally pure sands in the test is the collected uncontaminated sands.
(3) Figure. 2 should be modified to highlight the hydrocarbon film on surface of mineral particles in contaminated sands.
(4) Figure 9 should be modified to be clearer.
(5) Figure. 11 should be redrawn to distinguish between the curve of natural and enriched quartz sand.
(6) It is suggested to describe whether the original structure (e.g. hydrocarbon films) of the contaminated sands has changed in the test.
Reviewer 3 Report
Dear Authors,
I send you my suggestions.
1. In Keywords the first letter must be capital
2. between keywords use semicolon
3. In abstract write more about Materials and Methods. You write a lot of things for the results and a little for Materials. Also, a sentence as a conclusion.
4. I think that the Introduction section is short
5. Where in the Results section? With methods? Please separate
Best regards
Molla Katerina
Author Response
Dear Katerina Molla, please see the attachment.

Reviewer 4 Report
This research paper focused on analyzing physical-mechanical and hydrophysical properties of sandy soils exposed to long-term hydrocarbon contamination. The research method and results analysis were reasonable. The detailed comments were shown as follows.
1. The abstract should be revised. “The author, They clearly show that” These descriptions need to be modified. The structure of abstract should be revised, such as the detailed results of the research should be summarized in abstract.
2. The introduction should be revised.
3. Some style of units should be revised.
4. The serial number of the conclusions should be revised. (1) or ① was suggested to be used.
Reviewer 5 Report
On the Manuscript ID: sustainability-2210406
Type of manuscript: Article
Title: Analysing physical-mechanical and hydrophysical properties of sandy soils exposed to long-term hydrocarbon contamination
Authors: Ivan Lange *, Pavel Kotiukov, Jana Lebedeva Submitted to section:
Sustainable Engineering and Science,
I understand that the novelty of the study lies in the experimental investigation of the physical-mechanical and hydrophysical properties of oil-contaminated sandy soils. The authors motivate their research theme by the need to expand urban development on lands that previously housed various facilities involving petroleum activities from storage to industrial processing.
Knowing the impact of hydrocarbon contamination on soil properties is therefore of great interest. I have some questions and comments for the authors:
1. In the Introduction section, authors must provide at least a list of changes in soil composition and its physical-mechanical and hydrophysical properties that have been reported in the reference literature [13-18]. This will help the reader to understand the gap that the authors fill with their study.
2. The authors talk about oil-contaminated sands, aggregates, and even "functional groups that make up petroleum hydrocarbons", increasing number of polar groups (-OH, -COOH), etc. which is not supported by information about the materials used in the experiment. In other words, the authors do NOT provide any information regarding the chemical composition of the oil contaminants, and this makes the discussion on the functional groups inappropriate. My suggestion to the authors is to provide the FTIR spectrum or other chemical analysis of the oil collected from the contaminated sand to support their claim regarding the functional groups.
3. The authors should mention the method used for the images in Figure 3 and the statement regarding "the presence of oxidized hydrocarbon films on the surface of mineral particles". Was it SEM-EDX, something else? Again, FTIR or RAMAN spectroscopy would have been necessary for this discussion.
In conclusion of my review, the paper is not suitable for publication in this form. A major revision is recommended. The authors must provide chemical analyzes to show the true nature of the contaminants and to make sense of their discussion of chemical structure and functional groups.

Reviewer 6 Report
Review
General Comments
The manuscript is the result of exhaustive tests in pure and contaminated sand samples in order to analyze the influence of oil hydrocarbons sorption in sand physical properties. Several parameters were analyzed such as density, filtration, permeability, stress among others. The results show a significative relationship between these properties and the oil hydrocarbon content, which is adequately depicted in the manuscript. English is good and figures are very self-explanatory. This is why I consider it worth of being published. However, the manuscript could be improved in some aspects. Despite of the well referenced and informative nature of the manuscript, the introduction might be too short. Later on, it is mentioned briefly some other works in the same context, and I think this could be part of the context of this manuscript if these works were introduced from the beginning. The research methods miss some basic information that it is specified in the comments below. Methodology is not totally clarified, and I think it is important since it would enhance replicability and citation rate of the paper. Since there are multiple tests involved in the work, a flow chart depicting this, would be very helpful. Finally, the discussion, despite of being very informative about chemical processes related to sorption of oil hydrocarbons, mention little about the actual impact from the results of this research. This empirical analysis results would be enhanced if it is contextualized in a social framework. For example, there is a clear reduction of density related with the oil hydrocarbons content, even shear displacement period in contaminated sand was for 3 to 4 times longer than pure sand. Is this a problem for urban and industry building planning? Is there a risk involved? A little bit of discussion about this matter could enhance the results relevancy.
Specific comments:
Since the manuscript did not have line numbers, I did my best to describe the location of every comment.
In abstract, modify “author” for “authors”.
In Research methods, is it possible to have a map or coordinates of the study area location? It would be very helpful to have a map depicting the sample coordinates. If the non-contaminated sands were taken from a completely different place, it needs to be clarified.
Also, every sample mass was the same?
Every contaminated sample had the exactly same oil hydrocarbons content? (17.5 g/kg)
Paragraph after Table 1: It would make more relevant to also express difference between sand density in percentages too in the way of “contaminated sand in dense structure is ##% compared to pure sand”.
Same paragraph, it is mentioned previous studies with similar results that should be referenced in the line.
Third paragraph after Table 1: This plot is basically a scatter plot. It could be “Results acquired allowed the analysis of the relationship between filtration coefficient and sand density.
Figure 5. “Scatter plots”
Figure 5. It is interesting that in dense structure, contaminated sand has lower permeability than pure sand. Maybe it is related with the microfilm? This could be discussed.
Figure 7 and 8, I don’t think it is a suitable idea to label displacement curves like that, it looks confusing. I suggest using a different color for σ. Example: red color for σ1, σ2 and σ3 for pure sand curves and black color for σ1, σ2 and σ3 from contaminated sand curves.
Round 2
Reviewer 1 Report
This paper has revised. I have no more comment.
Reviewer 3 Report
Dear Authors,
Congratulation for the good work
Best regards
Molla Katerina
Reviewer 5 Report
The authors have substantially improved the paper compared to the first version. From the perspective of the requirements that I expressed in my first review, by adding in the introduction the information from the specialized literature regarding the chemical composition of oil-contaminated soils and its changes together with those of physical nature, I consider that the discussions of the authors based on the results which refer to chemical structures now become justified. In my opinion, the work can be published in its current form.
Reviewer 6 Report
The author has been very open and willingful to every suggestion and clarification requested not only for me but for every reviewer. The manuscript has been improved in every aspect, therefore I recommend its publication.